# Improving a data mining based diagnostic support tool for rare diseases on the example of M. Fabry: Gender differences need to be taken into account

Philipp Hahn[1‡*], Werner Lechner[2], Rainer-Georg Siefen[1], Christina Lampe[3], Peter Nordbeck[4], Lorenz Grigull[5], Thomas Lücke[1]

1 University Children's Hospital, Ruhr-University Bochum, Bochum, Germany, 2 KIMedi GmbH, Ulm, Germany, 3 ZSEGI Centre for Rare Diseases of the University Hospital Gießen, Gießen, Germany, 4 FaZiT Fabry Centre for interdisciplinary therapy of the University Hospital Würzburg, Würzburg, Germany, 5 ZSEB Centre for rare diseases of the University Hospital Bonn, Bonn, Germany

‡ This publication is part of the doctoral thesis of Philipp Hahn.
* philipp.hahn@klinikum-bochum.de

## Abstract

### Background

Rare diseases often present with a variety of clinical symptoms and therefore are challenging to diagnose. Fabry disease is an x-linked rare metabolic disorder. The severity of symptoms is usually different in men and women. Since therapeutic options for Fabry disease exist, early diagnosis is important. An artificial intelligence (AI)-based diagnosis support algorithm for rare diseases has been developed in preliminary studies.

### Objective

Our aim was to extend and train the questionnaire-based AI, capable of distinguishing patients with from those without rare diseases, to achieve satisfactory sensitivity for the detection of a single rare disease, Fabry disease, taking into account gender differences in disease perception.

### Methods

We collected 33 complete datasets from patients with confirmed Fabry disease. These records contained answered AI questionnaires, general information on disease progression, demographic information and quality of life (QoL) measures. The AI was trained to distinguish patients with Fabry disease from patients with relevant differential diagnoses. Its performance was assayed using stratified eleven-fold cross-validation and ROC curve calculation. Variables influencing the performance of the AI were examined with linear regression and calculation of the coefficient of determination.

**Data availability statement:** All relevant data are within the manuscript and its Supporting Information files.

**Funding:** The author(s) received no specific funding for this work.

**Competing interests:** I have read the journal's policy and the authors of this manuscript have the following competing interests: LG and WL are co-founders of the commercial company, Improved Medical Diagnostics IMD GmbH. The authors are not employees of Improved Medical Diagnostics IMD GmbH, nor are there other relevant declarations relating to consultancy, patents, products in development or marketed products, etc. The co-foundation of these authors (LG, WL) does not alter our adherence to PLOS ONE policies on sharing data and materials. All other authors have declared that no competing interests exist. This does not alter our adherence to PLOS ONE policies on sharing data and materials.

**Abbreviations:** AI, artificial intelligence; CAD, computer-aided diagnosis; ERT, enzyme replacement therapy; FD, fabry disease; OMIM, online mendelian inheritance in man; QoL, quality of life; ROC, receiver operating characteristic; SF-36, Short-form-36 health survey.

## Result

We were able to show that a relatively small sample is sufficient to achieve a sensitivity of 88.12% for the presence of Fabry disease, taking into account gender-specific differences in the disease perception during the pre-diagnostic phase. No confounders of the tool's performance could be found in the data collected concerning the patients' quality of life and diagnostic history.

## Conclusion

This study illustrates on the example of Fabry disease that differences between female and male Fabry patients, not only in the expression of symptoms, but also with regard to disease perception, might be relevant influencing variables for improving the performance of AI-based diagnostic support tools for rare diseases.

## Introduction

Rare diseases affect about 30 million people in Europe [1]. Fabry disease (FD, OMIM #301500) is one of the most common of the about 70 lysosomal metabolic diseases known to date [2]. With an incidence of 1:8000–1:15000 [3,4] FD is considered a rare disease following the definition of the European Union (incidence <5:10000) [5–7].

FD is caused by an X-linked mutation of the GLA gene (OMIM #300644; HGNC:4296), which leads to a reduced activity of the lysosomal enzyme α-galactosidase A. This causes accumulation of globotriaosylceramide (Gb3) and its derivative globotriaosylsphingosine (lyso-Gb3) in all tissues [3]. The resulting cell damage particularly affects the nerves, vessels, kidneys and heart. Initial symptoms can occur in early childhood and often include neuropathic pain (especially burning pain of the extremities), angiokeratomas, gastrointestinal complaints and hypo- or anhidrosis [7]. Kidney damage, cardiac problems and cerebrovascular complications, which can lead to a considerable reduction in quality of life [8,9] as well as life expectancy, usually occur in adulthood. The severity of the symptoms depends on the residual activity of α-galactosidase A. Due to the x-linked inheritance of the disease, women usually have a higher residual activity of α-galactosidase and a milder manifestation of the disease, but can also show the entire spectrum of symptoms [9,10]. The average age of onset of symptoms is higher in women, with acroparaesthesia occurring on average from the age of 14 in male patients and around the age of 19 in female patients [11]. Transient ischemic attacks or strokes (affecting 25% of untreated patients) occur on average at the age of 54 years in women with Fabry disease and at the age of 34 years in men with FD [7,12].

While the clinical presentation of the disease is thus sex-specific, this study considers gender differences in a broader sense, reflecting not only biological differences but also how individuals experience disease within healthcare systems and society, where the concept of gender can influence diagnosis, treatment, and outcomes [13–16].

The determination of enzyme activity can be used to diagnose FD, but in women this can be randomly false-positive or false-negative due to X inactivation. Therefore, in suspected cases, a complete sequencing of the GLA gene should be performed to confirm the diagnosis [10].

While there is currently no curative therapy for FD, some therapeutic approaches exist: Enzyme replacement therapy (ERT) is approved in Germany since 2001 and for some patients there is a chaperone therapy available since 2016. These therapies improve the prognosis and have a positive effect on some somatic symptoms as well as on the quality of life of those affected [17–21].

Other therapeutic approaches are under development, including substrate reduction therapy, gene therapy and substitution of mRNA [21–23].

In any case, the earlier the treatment is started, the better the effect [21,24]. The ideal time for starting ERT must be determined individually, considering the side effects and the effort required for therapy. ERT is indicated at the latest when symptoms are present and in asymptomatic male patients with classic mutation from the age of eight years [9,25–27].

The rarity of the disease and its heterogeneous presentation often lead to late detection of FD. Therefore there is a need for diagnostic support.

## Development of the AI-based computer-diagnosis system

Advances in computer science and the development of high-performance computing in recent years have led to a revival of the approach, which has been pursued for decades, of supporting diagnosis through applications of artificial intelligence [28–31].

Grigull et al. developed the AI-based computer-aided diagnosis (CAD) system investigated in this study, which was designed to recognize the response patterns of patients with rare diseases and distinguish them from patients with chronic diseases, non-rare diseases and psychosomatic diseases by processing a 53-item multiple-choice questionnaire and assigning scores to each of the output categories.

In large meta-analyses of the use of AI in rare diseases, Schaefer et al. 2020 and Visibelli et al. 2023 found that the most commonly used algorithms in CAD systems for rare diseases are Support Vector Machine, Random Forest and Artificial Neural Network. This reflects their ability to handle complex, high-dimensional data and their capacity to be trained on relatively small datasets to produce satisfactory results [32–35]. The AI examined in this study consists of exactly these three classifier systems, augmented by a fusion algorithm in accordance with the experience of A. Sieg et al., A. Rother et al. and the recommendations of Naydenova et al. [36–38].

The basis for the development of the questionnaire was the analysis of 20 structured interviews with patients suffering from a broad spectrum of rare diseases [39]. This spectrum corresponded to twenty disease entities with a high need for diagnostic support, including FD [40]. Patients with rare diseases often experience emotional stress during the period preceding a correct diagnosis. This stress is amplified when medical professionals do not believe them, when they undergo multiple or incorrect diagnoses, inappropriate treatments, or when they are mistakenly given a psychiatric or psychosomatic diagnosis [40]. The similarities in the experience of the pre-diagnostic phase and disease perception form the basis for the CAD system studied here.

The 53 questions of the questionnaire therefore target the everyday experience of the patients in search of a correct diagnosis under headings such as "Searching for causes", "Signs of illness", "Symptom control", "Being special", "Social environment" and "Everyday life". Age and gender are also recorded. Grigull et al. subsequently collected 1155 questionnaires completed by patients with a known diagnosis, 758 of which were from patients with at least one confirmed rare disease. This data set was used to train the CAD system to assign scores to patients for different categories and for evaluation. In this way, a sensitivity of 88.9% for the presence of an unspecified rare disease, 86.6% for a non-rare disease, 87.7% for chronic diseases and 84.2% for psychosomatic diseases was achieved [39].

## Objective

We hypothesized that the previously developed questionnaire-based AI could be trained to detect Fabry disease as well, considering differences in disease perception of female and male patients.

Additionally, we sought patient-related confounders for the performance of the tool using a supplemental query of the context of their diagnosis and a survey of their quality of life.

## Methods

To enable the CAD system to distinguish between Fabry patients and patients with a possible differential diagnosis, as required for this study, and to account for expected differences in the response patterns of male and female Fabry patients, we replaced the output categories of the individual classifiers and the fusion algorithm described above, which corresponded to the previous development stage of the system, with the categories 'Fabry woman', 'Fabry man' and 'Other'. The tool produced the higher value from 'Fabry Woman' and 'Fabry Man' and the value for "Other" as the output result. Participants were asked to answer the CAD questionnaire as they would have done at the time of their diagnosis.

Participants with FD were considered correctly identified by the AI if they received the higher score in the "Fabry" category.

The Short-Form-36 Health Survey (SF-36), well established for FD [8,41–43], was selected to assess quality of life as a potential confounder of the performance of the CAD system. It is a 36-item, multiple-choice questionnaire that measures different aspects of quality of life as scores on eight subscales. Four of these subscales are summarised by the physical sum scale. Scores can range from 0 to 100, with a score of 50 corresponding to the population average.

In order to record further confounders and to search for possible obstacles in the diagnostic process that had not been taken into account so far, a supplementary questionnaire was designed. This comprised a survey of master data (age, gender, educational level, occupation) and free-text questions about the first symptoms of Fabry disease and their time of onset. The known presence of Fabry disease in relatives or acquaintances at the time of diagnosis, the time of diagnosis and the specialty of the diagnosing medical practitioner were also recorded.

Using linear regression, we searched for correlations between the scores allocated by the CAD system and the supplementary information provided by participants with Fabry disease. Separate regression lines were calculated for women and men and the coefficient of determination $R^2$ was used as an expression of linear correlation [44].

Inclusion criteria for participation in our study were a molecular genetic diagnosis of Fabry disease and legal age of majority.

The targeted number of participants was between 25 and 35. Due to the rarity of the disease and especially in comparison to the sample sizes of individual disease entities in previous studies, this number of participants seemed achievable and appropriate in the survey period [39].

The recruitment period for this study commenced on 1st September 2019 and concluded on 1st April 2020. Participants were recruited by approaching patients and presenting the study during check-ups at the Fabry Centre Würzburg FaZit and the Centre for Rare Diseases Gießen ZSEGI, as well as during a local group meeting of the Morbus Fabry Selbsthilfe e.V. in Bochum in November 2019 and during the annual general meeting of the club in March 2020. During both events, the study was introduced in a short presentation and participation was made possible directly on site. Prior to their participation in the study, written consent was obtained from each individual. The study did not include minors.

As additionally seven women and six men with Fabry disease had visited the AI internet platform (https://diagnostik.kimedi.de) and answered the questionnaire since the last AI training [39], 46 data sets of Fabry patients were available at the beginning of the training phase (see Table 1).

Incomplete questionnaires could not be analyzed and would have resulted in the exclusion of the data set concerned. However, no such cases occurred. For the 13 data sets submitted online, the additional information on possible confounders was missing, so they could not be included in the confounder analysis.

**Table 1. Description of the study population.**

| | Participants female | Partici-pants male | Additional female Fabry patients | Additional male Fabry patients |
|---|---|---|---|---|
| Count | 24 | 9 | 7 | 6 |
| Average age (years) | 47,08 | 41,56 | 53,85 | 44,16 |
| Average age at symptom onset (years) | 19,58 | 7,44 | Unknown | Unknown |
| Diagnosis due to relatives with FD (%) | 50 | 33,3 | Unknown | Unknown |
| Average age at time of diagnosis (years) | 35,17 | 28 | Unknown | Unknown |

Based on the gender characteristic given in the questionnaire, each data set of participants was classified as Fabry woman or Fabry man.

In order to obtain a statement on the sensitivity of the AI with regard to Fabry disease, a reference group, named "Other", was formed for the training and the 11-fold cross-validation consisting of 42 data sets of patients with potential differential diagnoses to Fabry disease. The data sets were selected on 4th April 2022 from the data pool of the previous studies on this diagnostic support tool [38,39]. As all data sets had been anonymized before, it was impossible to trace them back to the patients during this study. Here, an attempt was made to represent as broad a spectrum as possible of the most important differential diagnoses identified by Hoffmann et al. 2009 [10]. We also aimed to achieve a similar average age and gender ratio in the Fabry group and the Other group.

Data sets of patients with several potential differential diagnoses were preferred. Thus, of the 82 differential diagnoses mentioned by Hoffmann et al., 27 could be represented in the Other group.

Those were: Rheumatic diseases (neuropathic and rheumatoid arthritis), fibromyalgia, systemic lupus erythematosus, Sjörgren's syndrome, uveitis, Osler's disease, dermatomyositis, diabetes mellitus, glomerulonephritis, Schönlein-Hennoch nephritis, M. Crohn's disease, ulcerative colitis, coeliac disease, irritable bowel syndrome, cluster (headache), migraine, multiple sclerosis, Guillain-Barré syndrome, neuropathy, ectodermal dysplasia, neurofibromatosis type 1, MELAS (mitochondrial encephalopathy, lactic acidosis, and stroke-like episodes) syndrome, Niemann-Pick disease, Pompe disease, haemochromatosis, porphyria and sarcoidosis.

Mean age and gender ratio were almost the same in both groups (see Table 2).

A total of 88 completed test questionnaires were thus available for training and testing the CAD (46 from patients with FD and 42 from patients with diseases that were significant for differential diagnosis).

Stratified eleven-fold cross-validation was used to evaluate the sensitivity of the AI. Stratified n-fold cross-validation is an established method for validating classifiers [45]. Kohavi et al. described an ideal validation with a stratified n-fold cross-validation, where n should take a value between 10 and 20. A cross-validation is described as stratified if the distribution of the sought characteristics in the test data set corresponds to the distribution in the training data set [45]. For the stratified elevenfold cross-validation, each individual data set was randomly assigned by a computer algorithm to one of eleven stratified data groups, so that each group consisted of the data sets from the CAD questionnaire responses of eight individuals. The term 'stratified' here means that the proportion of data from Fabry patients is the same in each of

**Table 2. Gender ratio and average age in Fabry group and Other group.**

| | Gender ratio women: men | Mean age |
|---|---|---|
| Fabry group | 2,29:1 | 43,3 years |
| Other group | 2,23:1 | 43,4 years |

 

these data groups, in this case approximately 50%. For each fold of the eleven-fold cross-validation, the data from ten of these eleven data groups were used to train the AI, after which the performance of the CAD was tested on the data group not used for training. This process was repeated a total of eleven times, each time using a different set of data for testing.

In this way, after eleven runs, each of the 88 data sets was used once to test the AI.

A Receiver Operating Characteristic (ROC) calculation was performed to determine diagnostic quality. ROC curves originate from broadcast technology and are used to both evaluate and compare test procedures. To calculate the ROC curve, value pairs for each patient were used, with their CAD output value for the Fabry category and their patient group ("Fabry" or "Other"). From this, a true positive rate (y-axis) and a false positive rate (x-axis) can be calculated for different thresholds (colour scale) and plotted on a coordinate system. As a guide, a diagonal line is drawn representing the shape of the ROC curve that would occur if the test procedure produced purely random results.

The Area Under the Curve (AUC) and the threshold value of the Youden Index are used for evaluation.

The Youden Index marks the point on the ROC curve that is closest to the values of an optimal test procedure. It can be identified as the point on the curve that is furthest from the diagonal line. The threshold value of the Youden Index is thus the cut-off score with greatest possible specificity and sensitivity. However, since the applied AI procedure provides the user with a score for both output categories, determining the best possible cut-off score is purely academic.

The AUC summarizes the performance of the test procedure across different cut-off scores. It is equivalent to the probability of the procedure to correctly classify a patient with FD, it can therefore be understood as sensitivity of the test procedure [46].

## Results

33 complete data sets with AI-, QoL- and supplementary questionnaire of patients with FD (24 women and 9 men) were collected. Furthermore, 7 female and 6 male patients participated via the online platform answering only the AI questionnaire.

When comparing the answer patterns of women and men, significant differences emerged. In 15 of the 53 questions, the most frequently selected answers by the two groups were diametrically opposed -women answered 'yes ', while men answered 'no ' or vice versa (see Table 3). Additionally, a clear difference was observed in question 16 ("Do you have the impression that your complaints/symptoms/phenomena are/were taken seriously by the physician(s)?", women answered on average "rather not", men on average "yes") and question 40 ("Do you have the impression that other people have to be considerate of you? ", women answered on average "no", men on average "rather so"). Small differences in the average response patterns were found in questions 1, 17, 21, 33, 34 and 35 (see Q53 questionnaire in supporting information). In the remaining 30 questions, the response patterns of men and women matched (see Fig 1).

### 11-fold cross-validation and ROC calculation

In each run of the stratified 11-fold cross-validation, all datasets were assigned to the correct output category, i.e., achieved a higher score in this category than in the opposite category. The ROC curve of the CAD system examined here shows an AUC value of 88.12%. The confidence interval (CI), in which 95% of the results lie, varies between 80.5% and 95.7%, which corresponds to a medium width.

The Youden Index threshold (i.e., the optimal cut-off score) is reached at 0.915. If the system were to apply a fixed threshold to determine the diagnosis, a value of 0.915 would indicate that patients are classified as having Fabry disease (FD) if the CAD system assigns them a score of at least 0.915. Under this condition, the system would achieve a true positive rate of 80% for the Fabry category and 83% for the Other category (see Fig 2). However, this threshold is not clinically relevant, as the CAD system categorizes patients by directly comparing the scores in the two categories (Fabry vs Other). This metric is primarily of interest for academic comparisons in future studies.

**Table 3. Questions with diametrically opposed most given answers of women and men (full questionnaire in supporting information) [39].**

| Question number | Question text | Most frequently selected answers |
|---|---|---|
| 4 | Is it difficult for you to put your complaints/ irritating phenomena into words? | women: yes men: no |
| 6 | Have you been submitted to plenty of investigations without conclusive results? | women: yes men: no |
| 8 | Have you seen different specialists for different complaints/irritating symptoms? | women: yes men: no |
| 13 | Have your complaints/ irritating symptoms repeatedly been given different names (diagnoses) over the course of time? | women: yes men: no |
| 14 | Did you search for doctors (specialists, experts) on your own initiative in the course of your diagnosis search? | women: yes men: no |
| 16 | Do you have the impression that your complaints/symptoms/phenomena are/were taken seriously by the physician(s)? | women: rather not men: yes |
| 18 | Have you ever reached the point where you have given up your search for a diagnosis? | women: yes men: no |
| 19 | Is it true that a psychological or psychosomatic disease was suspected? | women: yes men: no |
| 24 | Have you noticed any irritating peculiarities (e.g., discolouration of the skin, enlargement of body parts, trembling, twitching, etc.)? | women: no men: yes |
| 25 | Do you suffer from recurrent severe pain? | women: yes men: no |
| 30 | Is it true that you have been approached by people from your environment (family, acquaintances, friends, colleagues, etc.) on your physical abnormalities? | women: no men: yes |
| 36 | Were you considered unathletic as a child/adolescent (e.g., were you exempt from school sports or did not like to participate)? | women: no men: yes |
| 40 | Do you have the impression that other people have to be considerate of you? | women: no men: rather yes |
| 42 | Would you say that the ambiguity about the cause of your complaints/ irritating phenomena was the worst? | women: yes men: no |
| 44 | Do/did you have the impression that those around you (family, friends, acquaintances, colleagues, etc.) do not take your complaints seriously (e.g., someone says: "It' s not that bad...") | women: yes men: no |
| 46 | Is it true that you prefer staying at home (instead of going out/clubbing) since your complaints are obvious? | women: yes men: no |
| 51 | Do you use aids to help you cope better with everyday life? | women: no men: yes |

## Confounder and quality of life

The age range of the male participants answering the SF-36 ranged from 26 to 52 years. Their average score on the physical sum scale of the SF-36 was 49.4. Women in the same age group achieved an average score of 38.8 (see Fig 3). A score difference of 3–5 is to be understood as clinically relevant [47].

Neither for the age at the time of the first symptoms, the age at the time of diagnosis, for the presence of pain in the extremities at the time of diagnosis or for the delay between the first consultation and diagnosis, nor for the current quality of life or the level of education could regression lines with a higher coefficient of determination than $R2 = 0.17$ be calculated.

Thus, no influence on the performance of the CAD system can be assumed for these factors.

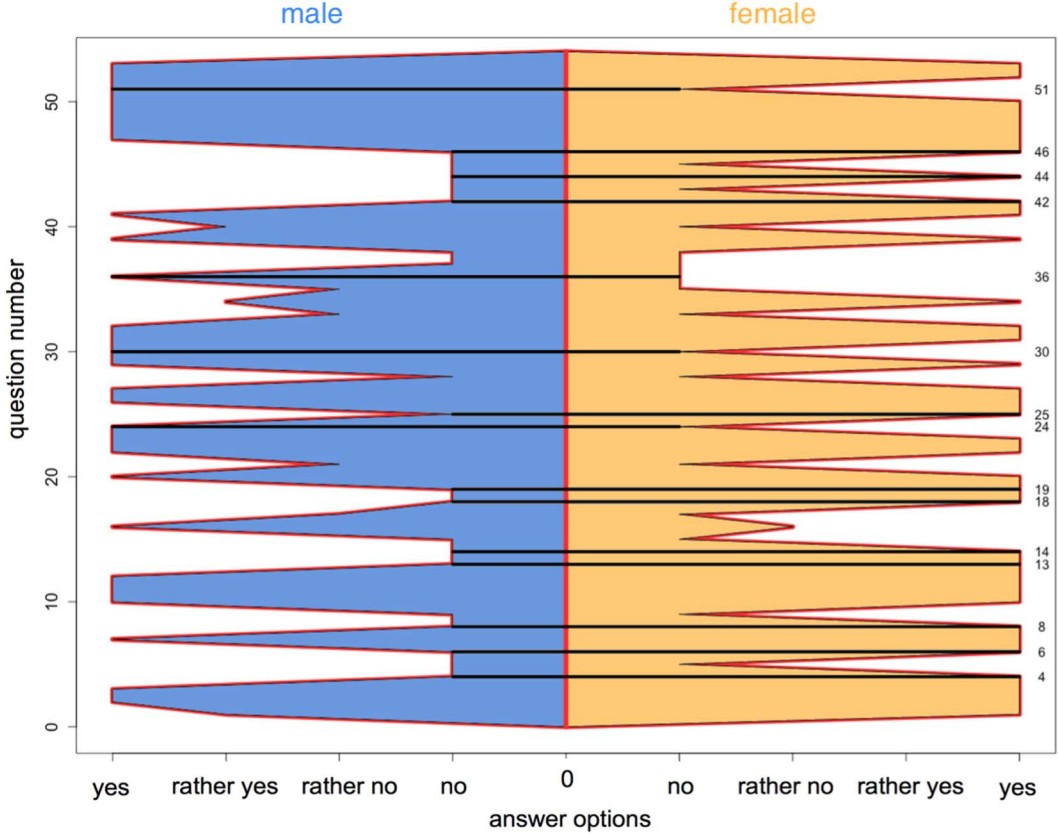

**Fig 1. Answer patterns of FD patients.** Answer patterns of men (left, blue) and women (right, yellow) with Fabry disease. Questions with diametrically opposed most frequently selected answers are marked with a black horizontal line.

## Discussion

- The CAD system tested here has a good sensitivity of 88.1% for the presence of FD and, for the first time, takes gender differences in disease burden into account.

- There are very few comparable published CAD approaches. To the extent that comparisons are possible, the performance of this system is satisfactory.

- Further studies with prospective testing and the extension of the test to other diseases are still pending.

Using ROC curve analysis, we demonstrated that the CAD system achieved a sensitivity of 88.1% in distinguishing Fabry disease from several common differential diagnoses. This performance was attained using a relatively small dataset, while accounting for differences in disease presentation and perception between affected men and women. Although a sensitivity exceeding 90% is typically desirable for diagnostic systems of this kind, the achieved value of 88.1% is notable given the limited dataset of only 46 FD patients.

The development of AI-based approaches for rare diseases is often based on small datasets, typically from 20 to 99 patients [33]. The sample size achieved in this study therefore seems suitable regarding the rarity of the disease and the size of the population.

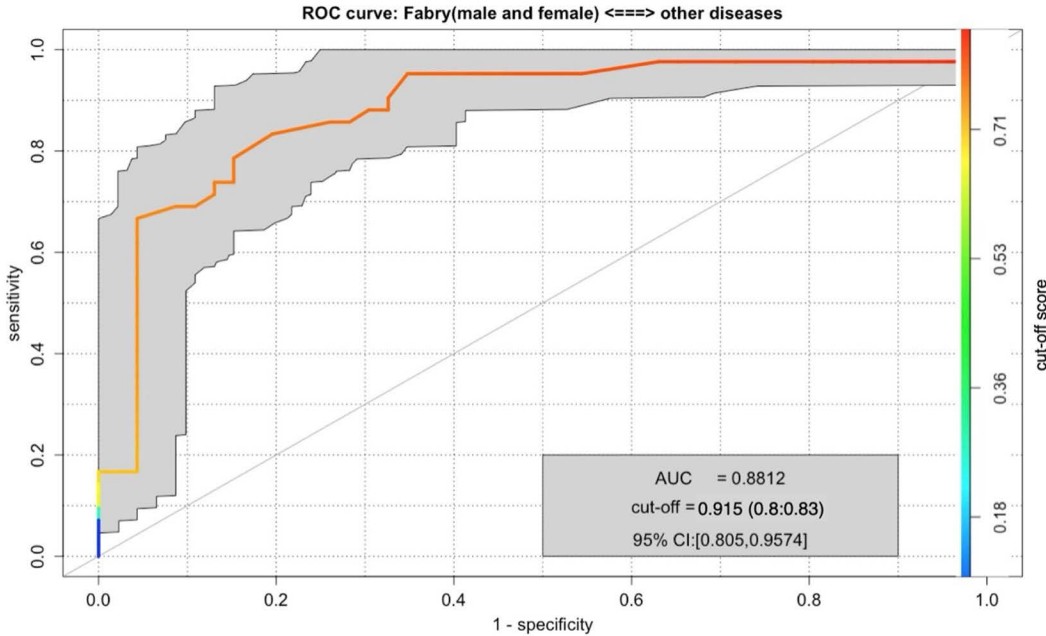

**Fig 2. ROC curve for Fabry disease (women and men) vs. Other.**

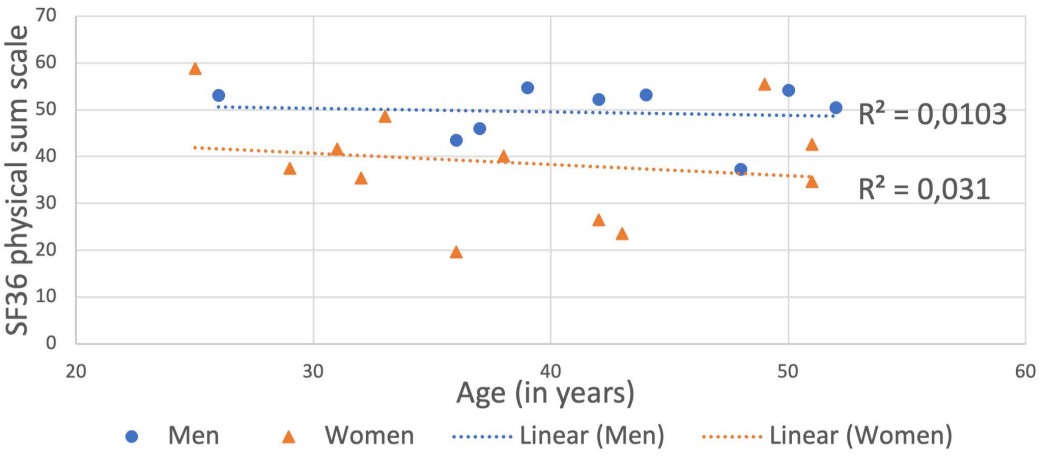

**Fig 3. Physical sum scale in SF 36 for participants aged 25–52.**

It is not possible to make a statement about the specificity of the system, as there is no information on the exclusion of Fabry disease in the patients of the control group, even though patients of the control group did not show FD typical symptoms.

The sensitivity of 88.1% for detecting FD is comparable to the similarly high sensitivity of 88.9% reported for the CAD system in identifying rare diseases in the preliminary 2019 study [39]. Notably, several of the differential diagnoses used as comparators in the present study are themselves rare diseases, underscoring the challenge of distinguishing between conditions within the broader category of rare diseases.

Visibelli et al. found 30 publications on AI applications for the diagnosis of rare diseases that were published between 2013 and the end of 2022. On average, these methods achieved an accuracy/AUC of 84.0% [32].

While there are few examples of questionnaire-based CAD systems for diagnosis in non-rare diseases [34,48], there is a lack of comparable publications on other questionnaire-based CAD systems used for rare diseases.

The questionnaire-based AI method published by Horowitz et al. in 2007 to support the diagnosis of gastroesopha- geal reflux was able to achieve a maximum sensitivity of 78% in the AUC analysis [34]. In 2011, Sun et al. presented a questionnaire-based AI method for CAD in sleep apnoea that achieved a maximum sensitivity of 88.0% [48]. It should be noted that these methods, in contrast to the one tested here, only used a single classifier and no fusion algorithm. There are currently no other, purely questionnaire-based AI systems for diagnostic support.

A project for diagnostic support using artificial intelligence in Parkinson's disease achieved a sensitivity of 95.58% in 2022 by combining two AI algorithms using a fusion algorithm [49]; this system was trained with a comparatively large data set of 195 speech recordings and, like the method presented here, evaluated using k-fold cross-validation and ROC analysis.

In 2012, Arning et al presented FabryScan, a screening tool for the early diagnosis of Fabry disease in patients with pain in the extremities. By analyzing a short questionnaire and three quantitative sensory tests, it achieved a sensitivity of 88% [50]. In addition, further diagnostic steps were necessary to confirm the suspected diagnosis, as is the case with the CAD system investigated in this study. The similarly strong performance of the latter approach is noteworthy, particularly given that it does not require a prior suspected diagnosis and involves relatively low user effort. However, the findings of the present study should be interpreted in light of several limitations, including its retrospective design and the limited number of questionnaires available for analysis.

For the first time in the development of this kind of CAD particularly regarding the differing ways in which men and women experience and cope with illness, as highlighted by findings from gender-sensitive medicine. This includes con- sideration of disease-specific factors that may influence symptom perception, health behavior, and diagnostic pathways [13–16]. The system was trained in such a way that the correct assignment as FD would have been accepted even if the gender-related expression type had not matched the gender of the participating test patient; after all, the system should refer to the correct diagnosis and not determine the gender of the person being examined. However, no such incongru- ence occurred.

[51] Although the proportion of participants for whom the diagnosis was initiated due to an affected relative was similar for male and female participants (women 37.5%, men 33.3%), it is noticeable that men tended to answer "no" concern- ing questions related to the difficult search for a diagnosis (in particular questions 6, 8, 13, 14, 18, 42, see Table 3). The response patterns in questions 19 and 44, which relate to how seriously patients' complaints were taken by those around them and whether they were wrongly diagnosed with a psychosomatic disease, are consistent with the now not-so-new finding that gender-specific prejudices and medicine shaped by the male norm may hinder the diagnosis and appropriate treatment of female patients.

Considering the pathophysiological mechanisms of the x-linked disease it can be assumed that symptoms are more severe in male patients, which may explain their ease of communicating them (see table 3, question 4).

The fact that the question about significant pain (question 25) was nevertheless predominantly answered in negation by the male participants, but positively by the female participants, may be due to a random under-development of pain symp- toms in the male participants, which can be attributed to the sample size, but perhaps also to gender-specific differences in the perception of pain [15]. Questions 24, 30, 36 and 51, which were answered positively by the male participants, might also be explained by the participants' more severe symptoms, as well as the possibly higher expectations of male role stereotypes in terms of performance in school sports.

To conclude, the CAD system evaluated in this study is unique in several key aspects compared to previously devel- oped systems. It can suggest the presence of a rare disease without requiring the user to identify it as a suspected

diagnosis. Unlike traditional approaches, it relies not on clinical data but on a questionnaire assessing disease perception. It also incorporates gender-specific differences in disease perception and utilizes multiple classifiers combined through a fusion algorithm. Despite being based on a relatively small dataset, the system achieves a sensitivity of 88.1% for detecting Fabry disease, thereby offering valuable support in initiating a definitive diagnosis (enzyme activity and sequencing of the GLA gene).

Our study has several limitations. First of all, Fabry disease is an x-linked hereditary disease. There are differences in the expression of the disease in men and women. The definition of a CAD-internal female expression type of Fabry disease is therefore more self-evident than it is the case for diseases with sex-independent expression. Nevertheless, heterozygous women can also exibit significant disease burden and their clinical phenotype can differ considerably from men [43]. In general, the lower average physical sum score on the SF-36 that we found in female participants in comparison to male is in line with what has been described in women with Fabry disease [47]. But the score for the male participants is higher than expected compared to a previous study on the quality of life of treated Fabry patients (see Fig 3) [18]. It is therefore possible that gender differences in disease burden are more pronounced in our study cohort than in the broader population of individuals with Fabry disease. Our data also shows that older participants have a lower quality of life than younger ones. This fits with the observations of comparable studies [47]. Blohm et al 2012 found renal function to be an independent factor of QoL, with male FD patients having impaired kidney function and QoL on average [8]. We did not analyze renal function, so it is possible that male participants in this study had a higher than average kidney function.

At the same time, as we do not have any comparable pre-training data on the performance of the system on FD, especially not regarding its performance without the distinction of female and male patients, we can only assume rather than prove the influence of this distinction between men and women on the improvement in performance. However, fundamental differences in disease processing [16] and the clearly distinct response patterns observed in this study suggest that incorporating gender-specific disease models may also enhance the sensitivity of CAD systems for other diseases.

To fully substantiate the role of gender-specific stratification in improving diagnostic accuracy, future work should include explicit testing of two modelling approaches: models trained with pooled data regardless of gender, and models incorporating gender as a separate feature or developing gender-specific models. Direct comparison of the performance metrics between these approaches will be crucial to determine whether stratification significantly enhances early detection, particularly for women who historically experience greater diagnostic challenges.

A significantly higher number of female Fabry patients than male patients participated in the study. This may explain why the scores in the Fabry and the Other categories show a greater difference for women (women 0.86, men 0.75). This supports the assumption that future training, in particular with additional data sets of male Fabry patients, could further improve the sensitivity of the system.

In the context of clinical application and the objective of facilitating earlier diagnosis of FD, the composition of the patient cohort used to train the CAD system in this study warrants critical consideration. At the time of diagnosis, participants were, on average, relatively old—35 years for women and 28 years for men—and experienced a longer diagnostic delay than reported in the cohort described by Reisin et al. (2017) [51]. This indicates that the CAD system was primarily trained on individuals who had already undergone a prolonged diagnostic odyssey. As a result, its current design may limit its potential to identify FD at earlier stages, which is a central aim of AI-based diagnostic support systems.

Given the sex-specific diagnostic patterns observed in FD – where males often present earlier with more classic features, whereas females may present with more subtle or heterogeneous manifestations – there is an increased risk that the predictive ability of the model will be unequal between the sexes. In particular, several questionnaire items reflect experiences of misdiagnosis, psychological labelling or diagnostic abandonment, which were more frequently endorsed by women. Hope gives the fact that among the participants were also four female patients who received their diagnosis before the appearance of the first subjectively perceptible symptoms (by family screening or as an incidental finding by an ophthalmologist). These participants were identified by the CAD as FD patients and, with average scores of ~0.92 for

"Fabry" and ~0.05 for "Other", showed even better results than the overall average of the participants (~0.90 for "Fabry" and 0.07 for "Other"). We therefore believe the clinical application of the CAD has the potential to benefit patients without a long diagnostic journey, too.

Still, while our model offers a promising approach to capturing patient-reported characteristics of Fabry disease, the inclusion of these "diagnostic odyssey" questionnaire items, which capture the real-world burden of delayed diagnosis, in the development of the tool risks reinforcing a late-diagnosis profile and potentially limiting the sensitivity of the system to identify patients earlier in their disease course, particularly women. An effective diagnostic support tool should ideally focus on symptoms and clinical signs present early in disease progression rather than the sociomedical consequences of delayed recognition. All questions origin from interviews reflecting the time and symptoms prior to diagnosis. No question should therefore be regarded as "survey question". Nevertheless, diagnostic delay, symptom burden, psychosocial stressors, and individual resilience all influence the pattern of response—beyond the underlying disease itself. After all, the questions used here were developed from interviews with individuals affected by a wide range of rare diseases, who consistently reported similar experiences during the pre-diagnostic phase of their condition.

Additionally it has to be taken into account that all participants knew their diagnosis and some of them had been dealing with the disease for years. In particular, the recruitment of participants via the patient organization may have contributed to reaching particularly committed and well-informed participants. Their knowledge of the disease may have influenced how participants recalled and reported their symptoms, potentially introducing recall bias or confirmation bias. Specifically, individuals might have emphasized symptoms that they now recognize as characteristic of Fabry disease or minimized symptoms they retrospectively deem unrelated. This could result in a pattern of responses that differs from how individuals with undiagnosed Fabry disease would answer the same questions. Consequently, the CAD system's predictive performance may be optimized for individuals who have already internalized their diagnostic label, rather than reflecting the symptomatology of individuals in the pre-diagnostic phase.

To address this, we recommend in future work a prospective validation in undiagnosed populations, refinement of the questionnaire to prioritize symptom-based rather than journey-based items, and gender-specific calibration to ensure equitable diagnostic support. This approach will help ensure that the tool fulfills its goal of facilitating earlier and more accurate diagnosis across the full clinical spectrum of Fabry disease.

The excellent results in the 11-fold cross-validation can be interpreted as an indicator that the differences between the data sets of participants with Fabry disease and the Other group were particularly pronounced. Since the Other group was formed from data sets of patients with possible differential diagnoses to FD, we believe that the contrast between the two groups, which was quite clear for the CAD system, would have been difficult to recognize for a clinician tasked with the diagnosis. However, a cross-check by experienced diagnostic specialists is not methodologically feasible, since the data sets based on which the CAD system makes the classification do not contain any clinical information that can be used as a guide for conventional diagnostics.

At the same time, it becomes obvious that this is a so-called black box application with a decision-making process that is not comprehensible to the user. Although this applies to many AI-supported CAD systems, it runs contrary to the frequently formulated prerequisite for a broader acceptance of AI applications in everyday clinical practice [32,52,53].

The survey of the quality of life at the time of the study may only provide a limited indication of the quality of life at the time of the initial consultation or diagnosis, and thus the significance of the quality of life as a confounder for the performance of the CAD system may only be inadequately concluded from this parameter.

One shortcoming of the system analyzed here is that it is language and culture-bound and so far, mostly used in Germany, Austria and Switzerland. A questionnaire-based system also requires sufficient understanding of a written questionnaire.

Future iterations of the tool will have to re-evaluate the weighting or inclusion of questionnaire items that primarily capture the experience of diagnostic delay rather than disease-specific symptomatology. In particular, analyses stratified by gender will be essential to ensure that the tool performs equally well in men and women.

Future validation efforts should include prospective studies involving individuals suspected of Fabry disease but not yet diagnosed, to assess the tool's performance in a diagnostically naive population. Such validation would help determine the true clinical utility of the CAD system for early detection and reduce the risk of overestimating its diagnostic accuracy.

The results of this study give us hope that by applying this approach (taking into account the differences between men and women, questionnaire approach)) AI-based CAD systems can be developed and trained, that can make a valuable contribution to the diagnosis of rare diseases in the future.

We intend to train the system in follow-up studies to detect other rare diseases and will strive to provide a proof of concept through prospective application.

## Supporting information

**S1 File. Questionnaire of the CAD-System.**
(PDF)

**S2 File. Supplementary Questionnaire.**
(PDF)

**S3 File. Answer- and Score-Data.**
(XLSX)

## Author contributions

**Conceptualization:** Philipp Hahn, Rainer-Georg Siefen, Lorenz Grigull, Thomas Lücke.

**Data curation:** Philipp Hahn, Werner Lechner.

**Formal analysis:** Werner Lechner.

**Investigation:** Philipp Hahn, Werner Lechner, Christina Lampe, Peter Nordbeck.

**Methodology:** Philipp Hahn, Werner Lechner, Lorenz Grigull.

**Project administration:** Philipp Hahn, Thomas Lücke.

**Resources:** Werner Lechner, Lorenz Grigull.

**Software:** Werner Lechner.

**Supervision:** Lorenz Grigull, Thomas Lücke.

**Visualization:** Philipp Hahn, Werner Lechner.

**Writing – original draft:** Philipp Hahn.

**Writing – review & editing:** Philipp Hahn, Werner Lechner, Rainer-Georg Siefen, Christina Lampe, Peter Nordbeck, Lorenz Grigull, Thomas Lücke.

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
