## [Decision Letter · Decision Letter 0]

PONE-D-24-47701Improving a data mining based diagnostic support tool for rare diseases on the example of M. Fabry: Gender differences need to be taken into accountPLOS ONE

Dear Dr. Hahn,

Thank you for submitting your manuscript to PLOS ONE. After careful consideration, we feel that it has merit but does not fully meet PLOS ONE’s publication criteria as it currently stands. Therefore, we invite you to submit a revised version of the manuscript that addresses the points raised during the review process. Please address the comments from Reviewer#1 before submitting the revised version.

We look forward to receiving your revised manuscript.

Kind regards,

Naveen Joseph

Academic Editor

PLOS ONE

Journal Requirements:

“I have read the journal's policy and the authors of this manuscript have the following competing interests:

LG and WL are co-

founders of the commercial company, Improved Medical Diagnostics IMD GmbH. The authors are not employees of Improved Medical Diagnostics IMD GmbH, nor are there other relevant declarations relating to consultancy, patents, products in development or marketed products,

etc. The co-foundation of these authors (LG, WL) does not alter our adherence to PLOS ONE policies on sharing data and materials.

All other authors have declared that no competing interests exist.”

Reviewers' comments:

Reviewer's Responses to Questions

**Comments to the Author**

1. Is the manuscript technically sound, and do the data support the conclusions?

Reviewer #1: Yes

Reviewer #2: Yes

2. Has the statistical analysis been performed appropriately and rigorously? 

Reviewer #1: I Don't Know

Reviewer #2: Yes

3. Have the authors made all data underlying the findings in their manuscript fully available?

Reviewer #1: Yes

Reviewer #2: Yes

4. Is the manuscript presented in an intelligible fashion and written in standard English?

Reviewer #1: Yes

Reviewer #2: Yes

5. Review Comments to the Author

Reviewer #1: Hahn et al. present a computer-aided diagnosis system for the detection of Fabry disease that accounts for sex-specific differences between patients. The authors used questionnaire data to extend and train an existing AI system, originally designed to distinguish patients with rare diseases from those with chronic, non-rare, and psychosomatic diseases, to detect Fabry disease. The authors have taken the important step of distinguishing the disease perception of male and female patients in their classification model. This tool has the potential to assist in more timely and accurate diagnosis of a rare disease, which could allow for earlier therapeutic intervention and dramatically improve the quality of life for patients with Fabry disease.

Below I have listed several major concerns and minor suggestions that I believe should be addressed be addressed by the authors.

Major concerns:

-The authors have described a need for diagnostic assistance in Fabry disease, as early diagnosis is important for the successfully use of therapies in these patients. The authors acknowledge that all participants in the study knew their diagnosis, in some cases for years, and they may not have answered the questionnaire in the same way as they would have before their diagnosis. However, they do not discuss how this aspect of the training data may affect the utility of their model in diagnosing new patients with Fabry disease.

Additionally, given that many of the sex-specific differences in answers to the questionnaire, I am particularly concerned that the tool described here cannot effectively be used to aid in the diagnosis of new patients, particularly for women. Of the questions with diametrically opposed answers between men and women, many of the women’s answers indicate a long and difficult journey before their diagnosis. If this is the data used to train the model, can the model aid in the diagnosis of new patients if they have not also experienced things like getting different names (diagnosis) for symptoms over time (question 13), giving up on a search for a diagnosis (question 18), a psychological or psychosomatic disease being suspected (question 19). If this tool is meant to aid in earlier diagnosis of Fabry disease, I believe the authors should eliminate survey questions whose answers refer to a long or challenging diagnostic odyssey from their model.

If this is not feasible, the authors should at least add a more robust discussion of the potential impacts of the structure of the participant data on the ability of their model to aid in diagnosis. Several of the authors have expertise in the clinical diagnosis of rare diseases and their perspective on this aspect of the model will provide important context to their work.

Since several of the authors have extensive expertise in clinical diagnosis of rare diseases, I ask the authors to add a more robust discussion of the potential impacts of the structure of the participant data on the usefulness of their model in aiding diagnoses, both in general and for women and men individually, given not only the fact that participants already know they have the disease, but that many seem to have a long diagnostic odyssey and are likely to answer a number of the survey questions very differently from those early in their search for a diagnosis.

-The authors emphasize the importance of sex-specific differences in both the expression of symptoms and disease perception. It is logical to distinguish male and female participants in a study of an X-linked disorder. But the authors show no data to indicate that distinguishing between male and female participants improves the performance of their model. Indeed, on page 21 they state, “we can only surmise the influence of the distinction between men and women on the improvement in performance,” but again show no data about this improvement in performance. If they wish to make these assertions, the authors should compare the AI as currently trained to account for sex-specific differences to a model that does not separate male and female participants. How much does the separation of sex-specific differences improve the algorithm’s specificity in detecting Fabry disease?

Furthermore, given the disparity between the number of included female and male participants with Fabry disease in the training set, do the authors see significant differences in the sensitivity of the AI for detecting Fabry disease in women compared to men. Any significant differences would be important information if this model is to be used clinically as well as for informing future directions for improving the model.

-The description of the stratified cross-validation is unclear and confusing. Particularly this sentence on page 11, “In each of the eleven runs, other randomly selected eight of the 88 data sets were not used to train the AI but were used for subsequent testing.” It would be helpful to more clearly and explicitly describe the overall data set used for cross validation and the approach to the stratified selection.

-On page 20, the authors state that their algorithm can indicate a rare disease without a pre-formulated suspected diagnosis, however, it was trained and tested on individuals who already had a Fabry disease diagnosis. It is unclear to me that the authors can make this claim. If they can they need to more clearly explain why this claim is valid.

Minor concerns:

-The authors use the terms “gender” and “gender differences” but I believe the differences they are referring to are a result of differences between biological sexes in experiencing an X-linked disorder, and thus the terms “sex” and “sex-specific differences” should be used instead. Please make this change to reflect the more accurate terminology.

-Please provide a more detailed explanation of the “physical sum scale.” From the context provided in the text I believe it quantifies the answers to the quality of life survey, but this should be explicitly described.

-In figure 1, please include a label or legend in the figure to make it easy to know which color denotes answers from male participants and which are from female participants to improve readability of the figure.

-Please provide more detailed figure legends. For example, in Figure 1, there are some horizontal black lines which have unclear significance. If these lines do not have significance, please remove them from the figure. For Figure 2, please provide more plain English detail of the comparisons or calculations done to generate the ROC curve.

-There is a typo on page 16: “KI-based approaches for rare diseases” should be “AI-based approaches for rare diseases.”

-The discussion of the commonly used algorithms in CAD systems for rare diseases does not add to the discussion section of this manuscript. This paragraph seems to address the reasoning behind why the classifier systems used in this algorithm were selected and seems better suited to an introduction section.

Reviewer #2: Very good research paper regarding a CAD support tool for Fabri disease including gender differences.

All considerations are present, including ethics, limits on specificity of the system possibly leading to over diagnosis. The "blackbox" aspect of this application is mentioned. The simplicity of the symptoms questionnaire is a strength to fit to most patients' in real life or care centres.

6. PLOS authors have the option to publish the peer review history of their article (what does this mean? ). If published, this will include your full peer review and any attached files.

**Do you want your identity to be public for this peer review?** For information about this choice, including consent withdrawal, please see our Privacy Policy .

Reviewer #1: No

Reviewer #2: **Yes: ** Dominique Pougheon Bertrand

---

## [Author Response · Author response to Decision Letter 1]

13 May 2025

We have gladly provided a full and detailed response to the reviewer and editor comments in the document entitled "Response to Reviewers".

---

## [Decision Letter · Decision Letter 1]

Improving a data mining based diagnostic support tool for rare diseases on the example of M. Fabry: Gender differences need to be taken into account

PONE-D-24-47701R1

Dear Dr. Hahn,

We’re pleased to inform you that your manuscript has been judged scientifically suitable for publication and will be formally accepted for publication once it meets all outstanding technical requirements.

Kind regards,

Naveen Joseph

Academic Editor

PLOS ONE

Reviewers' comments:

Reviewer's Responses to Questions

**Comments to the Author**

1. If the authors have adequately addressed your comments raised in a previous round of review and you feel that this manuscript is now acceptable for publication, you may indicate that here to bypass the “Comments to the Author” section, enter your conflict of interest statement in the “Confidential to Editor” section, and submit your "Accept" recommendation.

Reviewer #1: All comments have been addressed

2. Is the manuscript technically sound, and do the data support the conclusions?

Reviewer #1: Yes

3. Has the statistical analysis been performed appropriately and rigorously? 

Reviewer #1: Yes

4. Have the authors made all data underlying the findings in their manuscript fully available?

Reviewer #1: Yes

5. Is the manuscript presented in an intelligible fashion and written in standard English?

Reviewer #1: Yes

6. Review Comments to the Author

Reviewer #1: I thank the authors for their careful consideration and nuanced responses to the reviewers' comments. I believe the changes made have fully addressed all of the reviewers' concerns, make the paper stronger, and that this work should be accepted for publication in PLOS One.

7. PLOS authors have the option to publish the peer review history of their article (what does this mean? ). If published, this will include your full peer review and any attached files.

**Do you want your identity to be public for this peer review?** For information about this choice, including consent withdrawal, please see our Privacy Policy .

Reviewer #1: **Yes: ** Alexandra J. Scott

---

## [Editor Report · Acceptance letter]

PONE-D-24-47701R1

PLOS ONE

Dear Dr. Hahn,

I'm pleased to inform you that your manuscript has been deemed suitable for publication in PLOS ONE. Congratulations! Your manuscript is now being handed over to our production team.

Kind regards,

on behalf of

Dr. Naveen Joseph

Academic Editor

PLOS ONE